# HUMAN ACTION RECOGNITION BASED ON SPATIAL-TEMPORAL ATTENTION

## ABSTRACT

Many state-of-the-art methods of recognizing human action are based on attention mechanism, which shows the importance of attention mechanism in action recognition. With the rapid development of neural networks, human action recognition has been achieved great improvement by using convolutional neural networks (C-NN) or recurrent neural networks (RNN). In this paper, we propose a model based on spatial-temporal attention weighted LSTM. This model pays attention to the key part in each video frame, and also focuses on the important frames in each video sequence, thus the most important theme for our model is how to find out the key point spatially and the key frames temporally. We show a feasible architecture which can solve those two problems effectively and achieve a satisfactory result. Our model is trained and tested on three datasets including UCF-11, UCF-101, and HMDB51. Those results demonstrate a high performance of our model in human action recognition.

## 1 INTRODUCTION

Human action recognition is focused by more and more people in recent years, researchers have developed many different methods of human action recognition. For example, there are some models based on CNN (Carreira & Zisserman, 2017), independent subspace analysis (Le et al., 2011), Restricted Boltzmann machine (RBM) (Nie et al., 2015)and RNN (Zhu et al., 2016). Some of them achieve pretty great performances in human action recognition.The methods in human action recognition can be split into two main categories. One is the traditional machine learning, another is the deep learning.

As for traditional machine learning, the improved Dense Trajectories (iDT) (Wang & Schmid, 2013) method performs pretty well before the rapid development of deep learning. Because of the low speed of the iDT, there are some improved methods based on iDT. Stacked Fisher Vectors (SVF) (Peng et al., 2014) is thought as one of the best improvement of IDT. Actually, many new deep learning methods of action recognition would use iDT as one part of their networks to optimize their models. As for deep learning, there are lots of methods proposed recently which are classical and useful. There are three main deep learning methods which are based on CNN , RNN or both of them.

With respect to CNN, in order to remind the temporal messages in videos, 3D-CNN (Ji et al., 2013) has been obtained good performance which adds the temporal information to CNN. Long-term Temporal Convolutions networks (Varol et al., 2018) is demonstrated with increased temporal extents, it improves the performance in action recognition. Two-stream CNN (Feichtenhofer et al., 2016b) is made up of two deep networks, one is used as temporal networks and another is used as spatial networks. Pose-based CNN (Chéron et al., 2015) uses RGB frames and optical flow to recognize actions. Two-stream Inflated 3D ConvNet(I3D) (Carreira & Zisserman, 2017) gives a state-of-art performance in human action recognition, it uses a more advanced architecture, the model is trained on bigger data sets. Based on I3D model, Choutas et al. (2018) proposes a new methods which is potion and motion representation, this improves the performance when they combine it with I3D. As the development of CNN on human action recognition, many more complex and efficient architectures have been come up with.

The pure RNN-based models are usually used on skeleton data. This kind of data do not need CNN to extract features, just because the human in these videos is usually abstracted to the skeleton. There

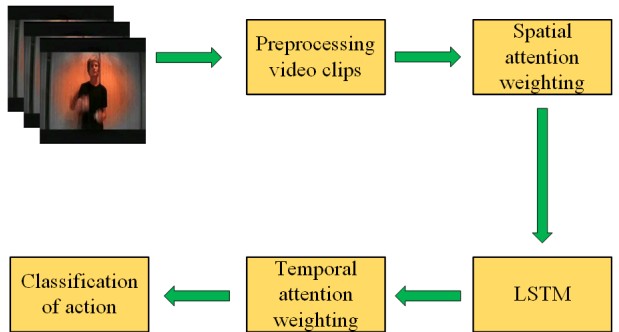

Figure 1: The architecture of our spatial-temporal model. We preprocess each video by divided it into images and extract features by putting those images into CNN. Then the model calculate the spatial attention and weights the feature cube. After this step, the weighted feature cube will be put into the main networks which are LSTMs. Via LSTMs, the model gets the outputs of the hidden layer.Then in order to gain the final result, temporal attention will be produced and weights those outputs. In the end, our model will give the classification of action in the video.

are many methods based on RNN for action recognition in skeleton data. Hierarchical Recurrent Neural Network (Du et al., 2015) divides human skeleton into five parts, then it uses RNN as the main networks in action recognition. Zhu et al. (2016) use Regularized Deep LSTM Networks to learn co-occurrence features of skeleton data. Part-Aware LSTM (Shahroudy et al., 2016) also divides human body joints into five groups in skeleton videos. Song et al. (2017) apply spatial and temporal attention mechanism in those skeleton videos, and the spatial attention on their models can be thought as the weight of human joints. They propose a method to calculate the temporal attention in skeleton data sets, and in this paper we will use it as the temporal attention of our model.

Other popular method uses RNNs as the main networks in RGB videos. However, as we all know it, CNN has achieved great performance on human action recognition both on images and videos (Carreira & Zisserman, 2017) (Feichtenhofer et al., 2016a), because it has great capability in dealing with images. So in this kind of model, CNNs are used to preprocess the video clips by extracting the convolutional features. The Long-term recurrent convolutional networks model (Donahue et al., 2015) is a classical architecture, it combines RNNs and CNNs organically. Based on CNN and RNN, Sharma et al. (2016) propounds a better model with visual attention mechanism, this model can focus on the important parts in each frame, it improves the accuracy of recognition. Lattice-LSTM (Sun et al., 2017) is put forward for action recognition by learning independent hidden state transitions of memory cells for individual spatial locations.

Although RNN is good at sequences, researchers think that's not enough. So attention mechanism (Vaswani et al., 2017) is proposed, it can improve the performance of RNN networks.The attention in videos includes spatial attention and temporal attention. The spatial attention in each frame shows the saliency of every part, but it's not very easy to generate attention automatically in images. Inspired by the success of attention in machine translation, Xu et al. (2015) come up with the attention mechanism including soft attention and hard attention in single still image, then their model generates the caption of the image by using soft or hard attention. But their attention mechanism still only works in still images until Sharma et al. (2016) apply their soft attention mechanism to human action recognition in videos. In consideration of many useless frames that have nothing to do with the action in a video, it's really necessary to select the key frames and clean the useless frames out. Then computers will be able to spend less resources and get better results. Temporal Segment Networks (Wang et al., 2016) combines a sparse temporal sampling strategy and video-level supervision, this model is efficient and effective. The temporal attention mechanism in skeleton data (Song et al., 2017)pays different levels of attention to the outputs of different frames. In this paper, for the purpose that our model can focus on the important part in the frame and pay more attention to more relevant frames in videos, based on the soft attention mechanism (Xu et al., 2015) (Sharma et al., 2016), we combine the temporal attention mechanism (Song et al., 2017) which has used in skeleton videos into our models. By using LSTM and spatial-temporal attention mechanism, we obtain satisfactory results in UCF11, UCF101, and HMDB51. In detail, in this paper, we choose Vgg19

(Simonyan & Zisserman, 2014) which is pretrained on the ImageNet (Deng et al., 2009) to extract the features in video frames, then we use LSTM as our models' main networks. The architecture of our model is illustrated in Fig. 1. It shows the progress of how our model dealing with the data.

In the summary, we make three contributions in this paper as follows:

- We propose a model with spatial and temporal attention in human action recognition in RGB videos. In our model, the attention of two dimensionality is combined with each other successfully.
- Compare with the baseline, we achieve the improvement of accuracy, which is about 7% in UCF11 and 15% HMDB51. Besides, our results is satisfactory when compared with other methods.

In our paper, in section 2, we will expound our method in detail. Then we plan to show the details of the experiment and also the results in section 3. In the end of this paper, we make a conclusion of our model in section 4.

## 2 SPATIAL-TEMPORAL ATTENTION MECHANISM

In this section, we will demonstrate our model in detail. First we are going to introduce how our model encode the videos. Then we will present how our model figure out the spatial attention in each frame and select the key frames in the videos, later we will show the progress of the decoder and the production of results. Finally we plan to introduce the loss function chosen in our model.

### 2.1 ENCODER OF THE MODEL

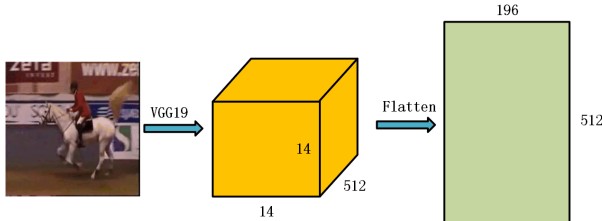

Figure 2: The schematic diagram of extracting feature. We use Vgg19 to extract the feature from the RGB images of the video. After getting the feature cubes, we flatten it into 2D matrix.

If we put the RGB video frames into the LSTM straightly, we would not get a satisfactory results. So we decide to encode these frames by using CNN like many other models. As showed in Fig. 2, in our model, we use the Vgg19 network pre-trained in Imagenet to extract the features in each video frame. After this step, we will get the feature cube which has the shape $L \times L \times D(14 \times 14 \times 512)$, for the purpose of easier calculation in next step, we flat the feature cube from 3D to 2D of shape $L^2 \times D(196 \times 512)$. The flat feature cube can be expressed as follows,

$$X_t = [x_{t,1}, x_{t,2}, ..., x_{t,i}, ...] \tag{1}$$

where the $t$ denotes the time which means the frame number in the videos, for example, if we extract twenty images in each video evenly, that means $t$ should be in the range of 1 to 20. The $i$ means the localization in each single image which has been divided into $14 \times 14$ parts, so the $i$ should be in the range of 1 to 196.

### 2.2 SPATIAL ATTENTION MECHANISM

The overall architecture is showed by Fig.3, in the left of the figure, $\alpha = \{\alpha_1, \alpha_2, ..., \alpha_t, ...\}$ donates the attention in the frames in each time $t$(the range of $t$ is 1 to T), so $\alpha_t$ is a matrix of shape 196 which is $L^2$. In this paper, we adopt the soft attention mechanism that is demonstrated by Xu et al.

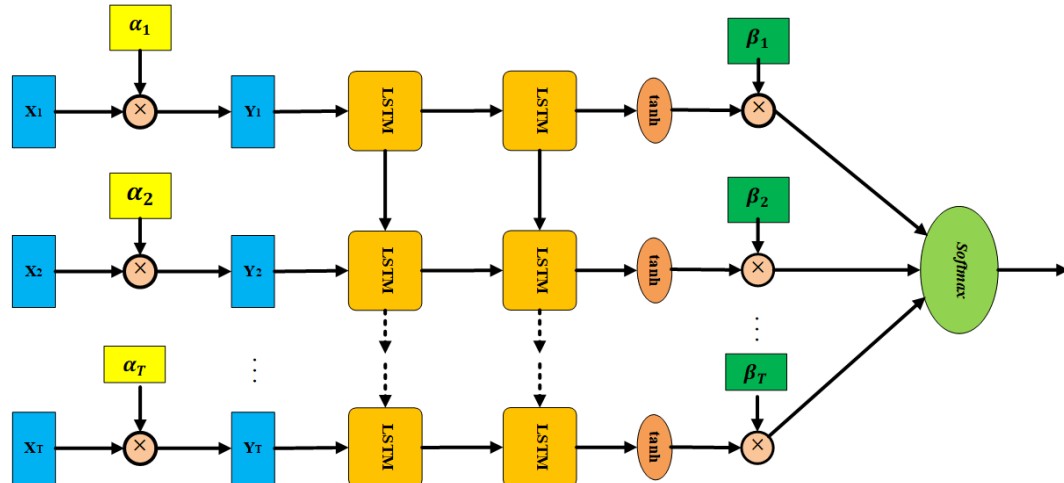

Figure 3: The detailed architecture of the model. This figure shows the overall architecture of our model, and there are three main parts in the model, which are spatial attention weight part , LSTM networks and temporal attention weight part.

(2015) and Sharma et al. (2016), which is shown in Fig. 4. The soft attention mechanism can be formulated as follows,

$$e_{t,i} = W_h^T \times h_{t-1} + W_x \times X_t + b$$
$$\alpha_{t,i} = \frac{\exp(e_{t,i})}{\sum_{j=1}^{L^2} \exp(e_{t,j})} \tag{2}$$

where the $h_{t-1}$ is the hidden state of LSTM in previous time $t-1$, and the $X_t$ is the feature matrix in present frame $t$. After the $e_{t,i}$ is calculated, the model will use the softmax function to compute $\alpha_{t,i}$, it is the attention weight in the frame $t$ and the ith parts of the frame. $\alpha_{t,i}$ is the probability that means the importance of the ith part for the action recognition in the frame. After computing the spatial attention, the model will weight the feature as the input of LSTM formulated in Eq. 3.

$$Y_t = \sum_{i=1}^{L^2} \alpha_{t,i} x_{t,i} \tag{3}$$

In previous section, we have already known the meaning of variable $x_{t,i}$, after the attention weights each parts in time $t$, $Y_t$ is computed as the input of LSTM in time $t$. This spatial weighting will make our model focus on the important parts in the frame rather than treat each part as the same materiality.

The initialization of the memory state and the hidden state of LSTM we adopt refers to the strategy used by Xu et al. (2015). It can be formulated as follows,

$$c_0 = f_{init,c}(\frac{1}{T} \sum_{t=1}^{T} (\frac{1}{L^2} \sum_{i=1}^{L^2} x_{t,i}))$$
$$h_0 = f_{init,h}(\frac{1}{T} \sum_{t=1}^{T} (\frac{1}{L^2} \sum_{i=1}^{L^2} x_{t,i})) \tag{4}$$

The $f_{init,c}()$ and $f_{init,h}()$ are both the multilayer perceptrons. T is the number of the total frames. The $c_0$ and $h_0$ will be used as the initial state of LSTM, in addition to this, these two values will be also used for calculated $\alpha_1$, the spatial attention of the first frame.

## 2.3 TEMPORAL ATTENTION MECHANISM

After putting the feature weighted by spatial attention through the LSTM, the soft attention model will get T classifications and count the number of each classification, then the final action classifica-

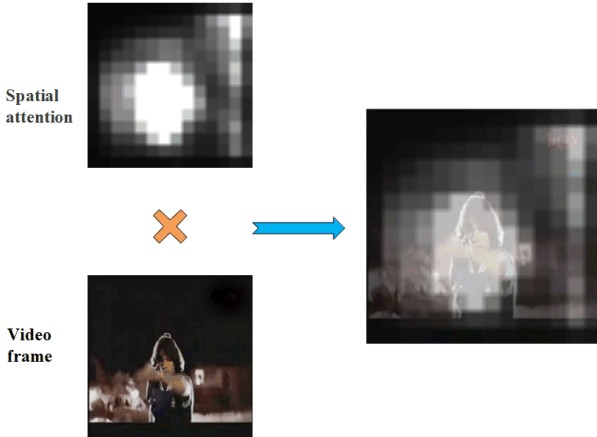

Figure 4: The schematic diagram of spatial attention. This is an Assumed effect of the soft attention mechanism in the video frame. The figure on the upper left is the visualization map of soft attention, the more lighting the part's color is, the more attention is pay to this part. So when combine with the corresponding video frame on the lower left which shows the scene that a man raise up his hand, the model should focus on these parts which in this frame should the man and his hand.

tion will be chosen by the maximum number, this kind of method seems like voting. However, in the video, if most of the frames are not relevant to the action, voting may be unreasonable. On the other hand, if the whole video frames is associated with the action, but most of them are repetitive, we may need only part of them to recognize the action. So we use temporal attention mechanism to make up for the soft attention mechanism. The temporal attention used in this paper can be illustrated as Fig. 5, it is proposed by Song et al. (2017).

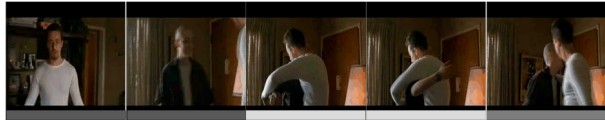

Figure 5: The schematic diagram of temporal attention. This picture describes how the temporal attention works on the model. For the sequence of the video frames in the figure, they demonstrate the action 'hugging', but only two of the five frames are relevant to 'hugging', so our model should select these two frames and pay more attention on it. The color under each frame is attached to the importance of the frames, more lighting denotes more importance.

In the right of Fig. 3, $\beta = \{\beta_1, \beta_2, ..., \beta_t, ...\}$ is the temporal attention used in our model. $\beta_t$ denotes the temporal attention in time $t$, it can be calculated by the equation as follows,

$$\beta_t = ReLU(W_h \times h_{t-1} + W_x \times X_t + b) \tag{5}$$

where the meaning of $h_{t-1}$ and $X_t$ has been introduced in section B, and we use the ReLU function which is the linear activation function to compute the temporal attention $\beta_t$. It's worth mentioning that the initialization of $h_{t-1}$ used here also uses the strategy demonstrated in Eq. 4. After this step, the model will weight the output in each time step of the LSTM as follows,

$$o = \sum_{t=1}^{T} \beta_t tanh(h_t) \tag{6}$$

The vector $o = (o_1, o_2, ..., o_C)$ will be used to calculate the final output, in which C is the total action classes of the video sets. At Each time $t$, the model will use the attention $\beta_t$ to weight the corresponded output of main LSTM networks in time $t$, and then sum this weighted vector as one vector $o$, then the softmax function will be used as follows,

$$\hat{z}_i = \frac{\exp(o_i)}{\sum_{j=1}^{C} \exp(o_j)} \tag{7}$$

In Eq. 7, $\hat{z}_i$ denotes the probability of the action belonging to the class $i$ in the video, the $i$ is in the range of 1 to C. So the final predicted class will judged based on $\hat{z} = (\hat{z}_1, \hat{z}_2, ..., \hat{z}_C)$, our model will choose the class of the maximum probability.

## 2.4 LOSS FUNCTION

In order train our model effectively, we formulate the loss function by using cross-entropy as follows,

$$L = -\sum_{i=1}^{C} z_i \log \hat{z}_i + \lambda_1 \sum_{i=1}^{L^2} \left(1 - \sum_{t=1}^{T} \alpha_{t,i}\right) + \lambda_2 \sum_{t=1}^{T} \|\beta_t\| \tag{8}$$

where the $z = (z_1, z_2, ..., z_C)$ is the one-hot vector, it is the groundtruth of the action. The $\hat{z} = (\hat{z}_1, \hat{z}_2, ..., \hat{z}_C)$ is the vector of probability that we have already talked before. $\lambda_1$ and $\lambda_2$ are the penalty coefficient of spatial and temporal attention which decide the contribution of them. The first term which uses cross-entropy makes our model predict more precisely. The second term is applied to force the model pay more spatial attention on more relevant parts in the frame automatically, then we also force $\sum_{t=1}^{T} \alpha_{t,i} = 1$. The third term is used to restrict the unlimited increasing of temporal attention.

## 3 EXPERIMENTAL

### 3.1 DATA SETS

In this paper, the proposed method is evaluated on three public data sets, which are UCF11, UCF101, and HMDB51.

**UCF11** (Liu et al., 2009): this data set is also called YouTube action. It is the RGB video set which contains 11 action categories. For each category, there are 25 groups with more than 4 clips in each group.

**UCF101** (Soomro et al., 2012): this data set is the expansion to UCF11. It contains 101 action categories, which can be divided into Human-Object Interaction, Body-Motion Only, Human-Human Interaction, Playing Musical Instruments, and Sports.

**HMDB51** (Kuehne et al., 2011): this data set contains 51 action categories with five types: General facial actions, Facial actions with object manipulation, General body movements, Body movements with object interaction, and Body movements for human interaction.

### 3.2 IMPLEMENT AND VISUALIZATION OF ATTENTION

The preprocessing of the model is transform the video clips into images. The images' shape is all reshaped to $224 \times 224$. Then we use the vgg19 which has been trained on the ImageNet to extract the feature from frame sequences. This step may produce very large feature data. Next, in this paper, we set the time steps of LSTM as 30 in UCF11, 20 or 10 in UCF101 and HMDB51. About the layers of LSTM, we have tried one layer or two layers, but it seems that both of them got very nearly accuracy, more layers does not have much difference. Besides, in order to avoid overfitting of our model in the training process, we use dropout layer. We trained our model on a GPU NVIDIA GTX 1080TI with the memory of 11GB.

When we train our model in HMDB51 data set by using stochastic gradient descent, we set the epoch as 500. We set the learning rate as $10^{-5}$, then the learning rate will be multiplied by $10^{-1}$ every ten epochs. Besides, in our experiment, we multiply the cross-entropy by 0.1, the value of $\lambda_1$ and $\lambda_2$ is set as 0.1 and 0.01.

The key point of our model is spatial and temporal attention mechanism, we will show the visualization of these two attention mechanism.

First, we will show the spatial attention of our model, which is the soft attention mechanism. As we have said in the previous section, our model will focus on the import parts in the frame automatically. In experiment, we get the results which correspond to our theory. Like the Fig. 6 illustrating, the

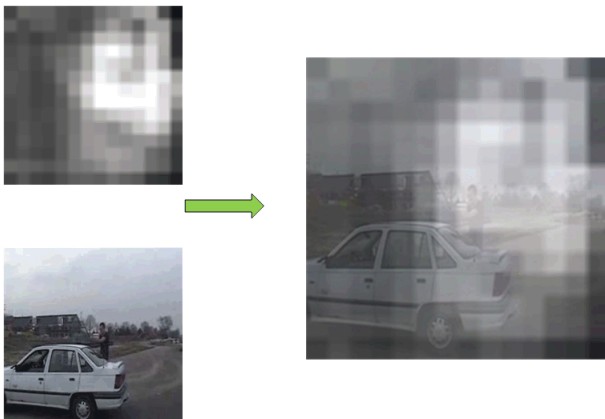

Figure 6: The spatial attention in the frame. This is the frame in the video clips which shows the action 'hit'. After putting the video frames into our model, we got the spatial attention weights of this frame, then we visualized the spatial attention. It's very obviously for us that our model focus on the man and his hand in the picture.

Table 1: The comparison with other methods in accuracy.We compare our model with other models including soft attention Sharma et al. (2016), ActionVLAD(RGB only) Girdhar et al. (2017), C3D(RGB only) Tran et al. (2015), Attention Pooling Girdhar & Ramanan (2017), Res3D Tran et al. (2017), Two-stream LSTM Yue-Hei Ng et al. (2015) and Videolstm Li et al. (2018)

| Model | Soft attention | ActionVLAD (RGB) | C3D (RGB) | Attention pooling | Res3D | Two-Stream +LSTM | Video lstm | Spatial-temporal (Ours) |
|---|---|---|---|---|---|---|---|---|
| UCF11 | 84.80% | - | - | - | - | - | - | **91.67%** |
| UCF101 | - | - | 85.20% | - | 88.00% | 88.60% | 88.90% | **90.25%** |
| HMDB51 | 41.30% | 49.80% | - | 52.20% | 54.90% | - | 56.40% | **56.77%** |

man hit the car by using a stick, and the picture in the upper left shows where our model focus on, the light place will get more attention while the dark place will get less attention, when combined with its corresponding frame, as we have expected, the model focus on the man and the stick on his hand automatically. That indicates the correctness of the spatial attention we've used in our model.

Then we will show the temporal attention in visualization as Fig. 7. The action 'brush hair', our model chooses different frames in different weights, and it also selects only some frames to recognize the action. When the woman raise up the comb, our model begin pay more attention on the frames. After the woman put down the comb, our model will give a bit attention on the frames. Besides, sometimes we can judge the action by only watching the first serval frame, so we have no necessary to watch the whole video. Our model seems have the ability to predict by only watching the first serval frame if the videos is very simple. This is a video clip from the HMDB51, our model final give the correct prediction 'brush hair'.

The visualization of the attention give us the proof that our model can select the important parts in the frame and the key frames in the clips automatically as we have talked before. The attention mechanism in the model came into play and get good results.

### 3.3 COMPARISON AND EVALUATION

Our model is trained and tested on those three data sets, it achieves the accuracy of 91.67% in UCF11, which improves about 7% compared to our baseline(soft attention model). Then in the data

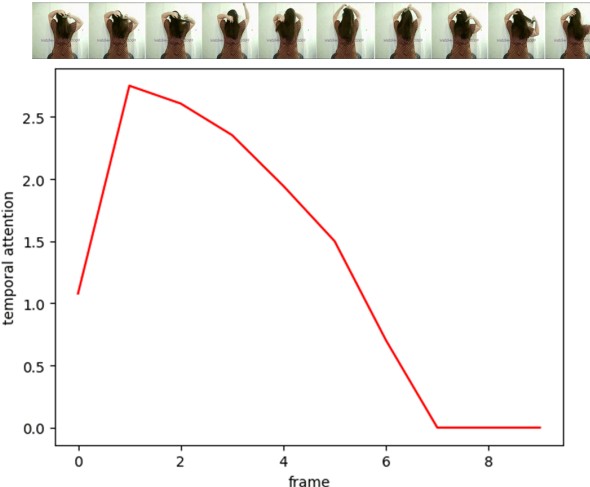

Figure 7: The temporal attention weight in frames. This is the temporal attention curve of the action 'brush hair'. Our model select the key frames that is relevant to the action.

set HMDB51, we get nearly 15% improvement which achieves the accuracy of 56.77%. In UCF101, we get the accuracy 90.25%. In UCF11 and HMDB51, our model both makes great improvement in the accuracy.

As for comparing with other model, we can see the comparison clearly in TABLE 1. In UCF101 data set, our model works so well and is better than some classic methods. Then in HMDB51 data set, it just reaches the accuracy 56% which is very ordinary compared to some state-of-the-art methods.

## 4   CONCLUSION

Human action recognition in RGB videos is a challenged job for us, but it's worth to do because of its great value. In this paper, we propose a human action recognition model based on the spatial and temporal attention mechanism. We choose the soft attention mechanism to select the relevant parts automatically in the frame, while use the temporal attention gates to pick up the key frames in the clips. Compared with the soft attention model which is the baseline of our model, our model makes great improvement in human action recognition in RGB videos. To explain the spatial-temporal attention more detailedly, we also visualize the spatial and temporal attention in the picture. In the end, we compare the accuracy of our model with other methods, our model performs very well.

However, there are still some weaknesses in our model. Firstly, this model is too simple, sometimes, the soft attention can not focus on the correct parts and the temporal attention can not select the key frame either. The soft attention mechanism and temporal attention can be replace by more advanced method or combined with other efficient models. So in the future, we plan to make our model more concentrated. We would improve the attention mechanism in spatial attention. About temporal attention, we can also make an improvement. Besides, when handling the complex data like HMDB51, our model doesn't perform very well. That means our model works not very well in the data sets which are very complex in the light and scene, in the next step, we should improve the performance on those kind of data sets.

## ACKNOWLEDGEMENT

This work was supported in part by the NSFC No. 91748208, No. 61876148, No. 61573268, and funded by China Postdoctoral Science Foundation of NO. 2018M631164 and the Fundamental Research Funds for the Central Universities of No. XJJ2018254.

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
