# OpenReview forum: "Human Action Recognition Based on Spatial-Temporal Attention"
_ICLR.cc/2019/Conference_

### Official Review · AnonReviewer2 · 2018-11-02
**Interesting results, but the paper is not matured enough**

**Rating:** 3
**Confidence:** 4

**Review:**

The paper propose an end-to-end combination of spatial and temporal attention for videos. The method first extracts a vgg19 representation to any frame, reduced with spatial soft-attention.  The attended vectors are then fed to LSTM with a soft temporal attention.

Strengths:

The problem of applying both temporal and spatial attention is important and challenging in general.

The reported numbers on HMDB51 and UCF101 are impressive, given the fact the authors only used rgb features. (Hope I haven't missed anything)


Weaknesses:

Recent datasets for action recognition, e.g., Moments in time, Charades, Youtube-8M etc, are missing. If the authors can show this model on any of these this will make their case stronger. I suspect the proposed model is limited to short-videos only,  keeping the spatial information means the features dimensions are increased by factor of 49. This is why Charades dataset can be very interesting to see, because the videos are longer. But also Moments in time, which is much larger.

The writeup should improve significantly: Grammar mistakes, typos e.g.,  donates/denotes, operations in equations (eq. 5,6), punctuation after equations. etc..

Even though the model is basic, It was really hard to follow. For instance, I couldn’t follow the whole discussion about T classifications, and the voting. What exactly are we classify? Another example, eq5, \beta_t is not an actual attention score, but the energy potential that we later use to calculate the attention in eq7.

Qualitative evaluation is barely there, one sample is not convincing enough, qualitative evaluation in vision-models should show many-many examples.  Fig 2,3. describe well-known techniques, I think it's better to add more examples instead.

To conclude:
This is important subject, but the paper is not matured enough. A better writeup, and evaluation on more recent dataset is necessary.

---

### Official Review · AnonReviewer1 · 2018-11-02
**Spatio-temporal attention weighted LSTM for action recognition is proposed. The novelty is low and the empirical evaluations are limited.**

**Rating:** 3
**Confidence:** 5

**Review:**

Summary:
The paper proposes a spatio-temporal attention weighting mechanism in LSTM, applied to the task of human action recognition. VGG19 based frame features are fed to LSTM, soft attention is calculated based on previous works and temporal attention is predicted using another small neural network. The features are weighted by these attentions and eventually the network is trained with a regularized cross entropy loss. Empirical results are given on three datasets for action recognition, UCF11, UCF101 and HMDB51.

Positives:
- The problem addressed is a relevant and challenging CV problem
- The idea of using of spatio-temporal attention is also interesting, as the actions are expected to have salient parts relatively sparsely located in space and time and focusing on them seems like an interesting direction to investigate.

Negatives:
- The paper is not well written in general
- The novelty is low as similar attention mechanisms have been used before. Papers have been cited in related works but differentiation in terms of what the current method adds is largely missing. The spatial attention is borrowed from Xu et al. (2015) and Sharma et al. (2016) and the temporal attention is relatively simple (similar ideas have been explored with CNNs as well eg. [A,B]) so the exact contribution and it's novelty is not convincing
- The results are not very convincing either, UCF11 is a very small dataset, on the bigger datasets the improvements over Video LSTM are small
- Self implemented baseline (the current implementation with same base CNN and LSTM networks without any spatial or temporal attention, \alpha=\beta=1 fixed) as well as ablation studies (what happens when only spatial or temporal attentions are used) should be added for assessing the contribution of the different components
- Some actual qualitative results should be added demonstrating the effectiveness of the proposed approach

[A] Kar et al., AdaScan: Adaptive Scan Pooling in Deep Convolutional Neural Networks for Human Action Recognition in Videos, CVPR 17
[B] Bilen et al., Action Recognition with Dynamic Image Networks, accepted for TPAMI 2018, arxiv 1612.00738

I feel that in the current form the paper is not ready for publication.

---

### Official Review · AnonReviewer3 · 2018-11-04
**Limited novelty and missing important experiments and comparisons**

**Rating:** 4
**Confidence:** 4

**Review:**

# 1. Summary
This paper presents a spatio-temporal attention LSTM for action recognition, where attention decides which pixels and frames are more important for classification. ConvNet features are extracted, a first layer of attention looks at the pixel level, then a second layer is applied at the temporal level. An LSTM is used to connect frame representation through time.

Weaknesses:
* The paper do not present substantial novelty compared to previous work. In fact, it has a strong overlap with (Song et al., 2017) (see #3)
* Some modeling choices are not well motivated (see #2)
* Ablation study showing that each modeling decision are motivated from a practical perspective is missing (see #4)
* The paper fails in comparing with relevant papers (see #4)


# 2. Clarity and Motivation
* Page 1 “many new deep learning methods of action recognition would use iDT as one part of their networks to optimize their models ”: this statement is not clear, please provide references of methods that do this. To my knowledge iDTs are part of the input of a ConvNet or used as complementary feature to other networks (e.g., C3D, I3D, …).
* Page 1, “Two-stream CNN (Feichtenhofer et al., 2016b)”: The reference might not really accurate. The citation should be: Two-Stream Convolutional Networks for Action Recognition in Videos Karen Simonyan, Andrew Zisserman.
*Page 1, “The pure RNN-based models are usually used on skeleton data“: it is not clear right away why the authors discuss some paper about skeleton data since it is not an application studied in this paper.
* Page 3, Sec. 2.2, “\alpha_t is a matrix”: why is it a matrix? It seems that it is a vector of length 196.
* By reshaping the features as Fig. 2, you loose the spatial consistency between neighbour pixels. How does Eq 2 deals with this? It seems that a better approach will be to have convolutions instead of fully-connected layers in Eq. 2. Have the authors considered this option?
* In neural machine translation models, usually the weights are normalized with a softmax before the weighted average of Eq. 3 and 6. It seems that here the alphas are normalized but the betas are not before Eq. 6. Any explanation about this?
* My comment above seems that is also related with the need of the regularisation term in Eq. 8. Probably it is not really needed in case that the betas are also normalized.
* A discussion is missing why the two attention models (spatial and temporal) are different. In principle, one could adopt the same kind of attention model for both the spatial and temporal component. One reason to have different models would be to consider the spatial relations between neighbour features in a frame (which is not the case of this model, as I highlighted above)
* Page 6, “The second term is applied to force the model to pay more spatial attention on more relevant parts in the frame automatically […]. The third term is used to restrict the unlimited increasing of temporal attention“: this sentence is a bit unclear. More details and intuition about how the 2nd and 3rd term work would be appreciated.


# 3. Novelty
From a methodological point of view the paper is not novel enough. It seems that the model is a combination between of the soft attention mechanism (Xu et al., 2015) (Sharma et al., 2016) and the temporal attention mechanism (Song et al., 2017). The overlap with the latter paper is substantial. From the application point of view, there is also little novelty given that the paper is tested on action recognition using relatively-small datasets.


# 4. Experimentation
Since this paper is an application paper, and there is not much novelty about the model, one would expect a comprehensive set of experiments.
* Ablation study is missing. Looking at the model in Fig. 3, it seems that not all the components are required. Some questions should be:
** Since attention is already working on the temporal domain, why do we need an LSTM model which seems redundant?
** What is the impact of removing the first layer of attention (\alpha)? And the temporal one (\beta)?
* The selected datasets are bit small scale. It would have been nice to see some results with bigger and more challenging datasets, such as Kinetics or similar.
* The paper fails in comparing with relevant papers (table 1). The topic of action recognition is widely explored, specifically for the datasets used in this paper. References and comparison numbers can be found here, for example: http://www.actionrecognition.net/files/dsetdetail.php?did=6;  and   http://www.actionrecognition.net/files/dsetdetail.php?did=5;

---

### Meta-Review · Area_Chair1 · 2018-12-13
**Strong agreement for rejection**

**Confidence:** 4
**Recommendation:** Reject

**Metareview:**

Average score of 3.33, highest score of 4.
The AC recommends rejection.